# Evidence for Contamination of Silica Microparticles in Advanced Platelet-Rich Fibrin Matrices Prepared Using Silica-Coated Plastic Tubes

**DOI:** 10.3390/biomedicines7020045

**Published:** 2019-06-19

**Authors:** Tetsuhiro Tsujino, Akira Takahashi, Sadahiro Yamaguchi, Taisuke Watanabe, Kazushige Isobe, Yutaka Kitamura, Takaaki Tanaka, Koh Nakata, Tomoyuki Kawase

**Affiliations:** 1Private Practice, Hiroshima 732-0066, Japan; tetsudds@gmail.com; 2Private Practice, Kawasaki 213-0033, Kanagawa, Japan; atakahashihdc@ybb.ne.jp; 3Private Practice, Ohta 373-0808, Japan; y-sada@mwd.biglobe.ne.jp; 4Division of Anatomy and Cell Biology of the Hard Tissue, Institute of Medicine and Dentistry, Niigata University, Niigata 951-8514, Japan; watatai@mui.biglobe.ne.jp; 5Tokyo Plastic Dental Society, Kita-ku, Tokyo 114-0002, Japan; kaz-iso@tc4.so-net.ne.jp; 6Department of Oral and Maxillofacial Surgery, Matsumoto Dental University, Shiojiri 399-0781, Japan; shinshu-osic@mbn.nifty.com; 7Department of Materials Science and Technology, Niigata University, Niigata 950-2181, Japan; tctanaka@eng.niigata-u.ac.jp; 8Bioscience Medical Research Center, Niigata University Medical and Dental Hospital, Niigata 951-8520, Japan; radical@med.niigata-u.ac.jp; 9Division of Oral Bioengineering, Institute of Medicine and Dentistry, Niigata University, Niigata 951-8514, Japan

**Keywords:** advanced platelet-rich fibrin, amorphous silica, coating, blood collection tube, health hazard

## Abstract

Platelet-rich fibrin (PRF) therapy has been widely applied in regenerative dentistry, and PRF preparation has been optimized to efficiently form fibrin clots using plain glass tubes. Currently, a shortage of commercially available glass tubes has forced PRF users to utilize silica-coated plastic tubes. However, most plastic tubes are approved by regulatory authorities only for diagnostic use and remain to be approved for PRF therapy. To clarify this issue, we quantified silica microparticles incorporated into the PRF matrix. Blood samples were collected into three different brands of silica-containing plastic tubes and were immediately centrifuged following the protocol for advanced-PRF (A-PRF). Advanced-PRF-like matrices were examined using a scanning electron microscope (SEM), and silica microparticles were quantified using a spectrophotometer. Each brand used silica microparticles of specific size and appearance. Regardless of tube brands and individual donors, significant, but not accidental, levels of silica microparticles were found to be incorporated into the A-PRF-like matrix, which will be consequently incorporated into the implantation sites. Presently, from the increasing data for cytotoxicity of amorphous silica, we cannot exclude the possibility that such A-PRF-like matrices negatively influence tissue regeneration through induction of inflammation. Further investigation should be performed to clarify such potential risks.

## 1. Introduction

Similar to platelet-rich plasma (PRP), platelet-rich fibrin (PRF) has been demonstrated to be effective in tissue repair/regeneration in various fields of regenerative medicine [1,2,3,4,5]. The major advantages of PRF over PRP are that it does not necessitate superior operator skills, anti-coagulants, or coagulation factors, implying that fewer sections of the PRF preparation protocol can be biased. Thus, the use of standardized blood-collection tubes and centrifuges under the same centrifugal conditions reproducibly prepare similar-quality PRF from the same individuals at the same time points. Therefore, PRF has been increasingly used around the world.

However, PRF users have recently faced a serious problem where, except for some brands specialized for PRF preparation, major medical equipment manufacturers have discontinued the production of plain glass tubes for blood collection [6]. Therefore, some PRF users have had to purchase PRF-specific glass tubes, even though conventional, non-approved glass tubes are less expensive than approved plastic tubes and can be obtained routinely from non-authorized distributors. If such glass tubes are not available, many PRF users would use silica-coated plastic tubes specific for blood coagulation, which were originally produced for diagnostic use, without a careful quality check. In fact, many manufacturers caution PRF users against the use of silica-coated plastic tubes, including plastic tubes containing silica-coated film, for the preparation of PRF for regenerative therapy because these products have not been approved for PRF preparation by regulatory agencies, unlike blood-transfusion bags in Japan and probably all other countries. Nonetheless, medical and dental clinicians in Japan still have the right to use their discretion while choosing blood collection tubes [6]. This regulatory matter is discussed in detail in the Discussion section.

Upon contact with the glass surface, coagulation factor XII in the plasma can be activated to consequently produce a fibrin clot through the activation of the intrinsic coagulation cascade [7]. Since the plastic surface does not have the potential to stimulate coagulation, PRF cannot be prepared in plastic tubes by the well-known centrifugation-based protocol. We have recently developed a method using a synthetic collagen-like protein to indirectly stimulate coagulation via activation of platelets and reproducibly prepare a PRF-like matrix [8]. In addition, in a preclinical animal study, we demonstrated that the combination of PRF and particles made of the collagen-like protein could be used as a safe and promising bone substitute in bone regenerative therapy.

However, such hybrid plastic tubes are not yet available commercially, and therefore many PRF users employ existing products as replacements of the original plain glass tubes. Among the existing products, the plastic tubes for blood coagulation meet their requirements [9]. Although detailed specifications seem to vary for individual products, they commonly contain silica microparticles inside the tubes because of the glass-like property of silica. However, many PRF users rarely pay attention to the possible health hazards. It has been reported that silica microparticles, depending on their amorphousness and size, can be harmful to our body [10,11]. As these products are generally used for diagnostic purposes only, to date, manufacturers have not disclosed the information regarding either specifications or ease of detachment of silica microparticles.

To evaluate the quality of the PRF prepared using silica-coated plastic tubes, we characterized the morphology of the silica microparticles and examined their possible contamination in the resulting PRF matrix.

## 2. Experimental Section

### 2.1. Preparation of Advanced-PRF (A-PRF) Matrices

After obtaining individual written informed consent, blood samples were collected, without anticoagulants, from six non-smoking healthy male volunteers aged 30–60 years. The study design and consent forms for all procedures (project identification code: 2297) were approved by the ethics committee for human participants at the Niigata University School of Medicine (Niigata, Japan) on 14 October 2015, in accordance with the Helsinki Declaration of 1964, as revised in 2013.

Fresh blood samples (approximately 9.0 mL) from each donor were collected into two brands of evacuated silica-coated plastic tubes, Vacuette^®^ serum clot activator tubes (Greiner Bio-One GmbH, Kremsmünster, Austria) and Neotube^®^ celite tubes (NP-PS0909; NIPRO Corp, Osaka, Japan) or evacuated tubes containing silica-coated film, Venoject II^®^ clot activator film-containing tubes (VP-P100K; TERUMO Corp, Tokyo, Japan) as a variation of silica-coated tubes. The specifications of these tube brands are summarized in Table 1. Blood was immediately centrifuged at 200× *g* for 14 min (A-PRF protocol) using a Duo centrifuge (Process, Nice, France) at ambient temperature (20–22 °C).

Quality checks were carried out on individual blood samples by performing platelet and other blood cell counts using a pocH 100iV automated hematology analyzer (Sysmex, Kobe, Japan).

### 2.2. Detachment of Silica Microparticles from the Inner Walls of Tubes or Films

Disinfectant, 70% ethanol (2 mL) was added to each tube which were then tapped several times and sonicated for 60 s in an ultrasonic cleaner (Citizen, Tokyo, Japan). The suspension of detached silica microparticles was transferred into plastic dishes and air-dried on a clean bench.

### 2.3. Scanning Electron Microscope (SEM) Examination

Dried silica microparticles were attached to carbon tape and directly examined under a scanning electron microscope (SEM) (TM-1000, Hitachi, Tokyo, Japan) with an accelerating voltage of 15 kV [12,13].

### 2.4. Spectrophotometric Quantification of Detached Silica Microparticles

Pure water (2 mL) was added to each tube and silica microparticles were detached thoroughly as described above. Then, the concentrations of silica suspensions (100 μL) were determined using a spectrophotometer (PiCOSCOPE; Ushio Inc., Tokyo, Japan). In a preliminary study, we confirmed that KOH did not significantly influence either the silica microparticles or the spectrophotometric determination.

### 2.5. Enzymatic Degradation of A-PRF Matrices to Release Silica Microparticles

A-PRF clots were gently compressed with a compression device (JMR, Niigata, Japan) [14], cut into 6–8 pieces, washed with Phosphate Buffered Saline (PBS) three times and degraded in 0.1% Trypsin + 1.06 mM Ethylenediaminetetraacetic acid (EDTA) (Life Technologies, Carlsbad, CA, USA) at 37 °C overnight until almost complete degradation occurred. Fibrin and cell-derived debris were further lysed in 1 M KOH for 1 h. The released silica microparticles were washed three times with pure water to remove colored elements, such as hemoglobin, and resuspended in pure water (2 mL). The concentrations of the silica microparticles were determined using a spectrophotometer.

### 2.6. Statistical Analysis

The data were expressed as mean ± standard deviation (SD). For two-group comparisons, statistical analyses were conducted to compare the mean values by Student *t*-test (SigmaPlot 12.5; Systat Software, Inc., San Jose, CA, USA). Differences with *p* values < 0.05 were considered significant.

## 3. Results

The macroscopic appearance of the silica-coated plastic tube (Neotube) pre- and post-centrifugation is shown in Figure 1. Before centrifugation, the tube wall appears homogenously hazy overall but some spots were dense. In contrast, after centrifugation, the tube wall turned out to be clear.

The components on the hazy wall of pre-centrifugation tubes were examined using SEM. Microscopic observations of silica microparticles contained in Neotube, Vacuette and Venoject II tubes are shown in Figure 2. After vigorously washing the inside of the tubes with 70% ethanol, the resulting suspensions were dried on plastic dishes and examined with SEM. According to manufacturers’ information, the inner walls or the plastic film are coated with silica. However, their size and shape varied with individual tube brands. Silica microparticles contained in the Venoject II tubes were composed mainly of small pebble-like crushed microparticles, while those of Neotube were diverse and larger. Neotube tubes contained pieces of broken honeycomb structures and needle-shape structures in addition to small pebble-like microparticles. Vacuette tubes were composed of both pebble-like, crushed microparticles and pieces of short cylinder-like structures. In general, microparticles contained in Venoject II tubes were the smallest among the silica microparticles examined.

A-PRF-like clots formed in individual plastic tubes after centrifugation and the balance of A-PRF-like matrices and red thrombi are shown in Figure 3. Compared with other PRF derivatives prepared by high-speed centrifugation, A-PRF clots prepared by low-speed centrifugation are usually attached with a longer red thrombus than that of high-speed centrifugation. However, when silica-containing tubes that efficiently facilitate coagulation are used, the red thrombus became longer and larger than glass tubes.

SEM observations of silica microparticles embedded in and attached to individual A-PRF-like matrices are shown in Figure 4. In all the matrices, silica microparticles were found. While silica microparticles were almost homogenously distributed in the A-PRF matrix prepared by Neotube, they were localized in several regions in the cases of Vacuette and Venoject II.

The contents and percentages of silica microparticles contained in A-PRF-like matrices versus whole contents of pre-use tubes are shown in Figure 5. We quantified the contents of silica microparticles using a spectrophotometer by modification of our previous method that was applied to platelet counts [15]. In a preliminary study, we confirmed a strong correlation between absorbance values and dilutions of silica suspensions as good standard curves (Figure 5a). Based on this data, the percentages of silica microparticles entrapped and retained in A-PRF-like matrices varied with individual tube brands. The highest recovery (approximately 30.9%) was observed in the A-PRF-like matrix prepared by Neotube. In contrast, the recovery in the cases of Vacuette and Venoject II were 5.2% and 10.6%, respectively, and much lower than that of Neotube.

## 4. Discussion

According to the manufacturers’ information [16], the silica microparticles used for coating the Neotubes are celite, which are amorphous silica. As for the silica microparticles used in other brands, we speculate that amorphous silica is probably used, based on the general consensus regarding the health hazards of crystalline silica [11,17] and/or the material safety data sheet provided by the other manufacturers [18,19]. However, we could not reach a conclusion based on this information.

The specific gravity of amorphous silica (2.15–2.30) is much higher than red blood cells (approximately 1.10); therefore, silica microparticles can be precipitated immediately after centrifugation under citrated conditions [20,21]. However, it is plausible that without anti-coagulants, coagulation to form a fibrin clot is triggered immediately by contact with silica microparticles and silica microparticles are entrapped within the fibrin clot. Our findings expectedly supported this scenario; however, for many PRF users who do not expect contamination of silica components in the PRF matrix, this may prove to be a surprise. Although the recovery rate may be influenced by the particle size and other factors, silica contamination was observed in all the plastic tubes tested. These findings showed that the PRF matrix prepared by silica-containing plastic tubes undoubtedly carries the silica microparticles into the implantation sites. These microparticles can be released like growth factors as fibrin clots are degraded by plasmin or other non-specific endogenous proteases and may influence the surrounding tissues and cells.

Despite contamination of silica microparticles in the resulting A-PRF-like matrices, the amounts varied with tube brands. The surface area of the silica coating film is substantially smaller than those on the inner wall of the tube. Thus, it is obvious that the mass of the silica microparticles in Venoject II tubes is much lower than that in other tube brands (Vacuette and Neotube) and consequently the mass of silica microparticles included in A-PRF-like matrix in Venoject II tubes must be lower than those of other brand tubes. In addition to the mass of silica microparticles, the silica entrapment and retention processes are primarily thought to be influenced by other factors, such as appearance and the size of the microparticles. Furthermore, these factors may influence the recovery during extraction and light transparency during spectrophotometric determination. It is plausible that as the microparticles become smaller, the entrapment and retention become more difficult. Therefore, we think that the reduction in recovery may not be necessarily related to or caused by the reduction in entrapment and/or retention.

On the other hand, even though amorphous silica is less toxic than crystalline silica and is actually used in a variety of products, including food and toothpaste, recent investigations have raised the possibility that amorphous silica can cause inflammation or cytotoxicity [22,23,24,25,26]. It is further described that particle size, which is usually correlated to the surface area, is more important than dose in exerting such negative effects. Since the cytotoxic and inflammatory effects of amorphous silica are thought to be mainly due to the reactive oxygen generated on the silica surface, a larger surface area per given mass is more reactive and has a higher potential to generate reactive oxygen species [22,23]. In addition, as described in the case of the complex interaction of crystalline silica [10], at present, we cannot exclude the possible interactions of amorphous silica at non-toxic levels with unidentified factors to exert unexpected health hazards.

In Japan, a new regulatory framework for regenerative medicine, including PRP/PRF therapy, was established in 2014, and individual hospitals and clinics are required to pass inspection by the regulatory agencies prior to treatment of their patients with PRP/PRF [6]. Nonetheless, medical and dental clinicians in Japan still have the right to choose the blood-collection tubes on the basis of their discretion for the treatment of patients [6]. Thus, the use of silica-coated plastic tubes is not prohibited for PRF preparation. Even though individual countries have individual regulatory frameworks for medical treatment, similar situations regarding PRF therapy have been found in several other countries. However, from a medical but not commercial point of view, we need to survey those individual situations at the global level and propose universal standards of medical devices, besides preparation of protocols, for patient-oriented PRF therapy through international collaboration.

The major advantages of PRF are low cost and high safety. Introducing possible silica hazards to this balance seems risky and may hamper the advances in PRF therapy. O’Connell forthrightly raised his concern over its use in PRF preparation in his article [27] and his concerns about safety have not yet been eradicated. To date, many efforts have been made to clarify the hazards of amorphous silica at cellular and molecular levels [10,23,24,26]. Until the safety of amorphous silica for implantation use is assured by international or national regulatory authorities, we should only use the conventional types of plain glass tubes for PRF preparation for the benefit of patients.

## 5. Conclusions

The most appropriate therapeutic methods are generally selected based on the balance between risk and benefit. PRF therapy alone is superior to other therapies using medicines or bone substitutes in terms of cost and safety. Thus, it can be confirmed that PRF therapy is advantageous based on the balance between low risk and low benefit. Until the possible health hazards of amorphous silica microparticles contaminated in PRF matrices cannot be ruled out by strong evidence, PRF therapy using silica-containing tubes has no convincing advantages over other costly therapies and only increases the risk of complications.

## Figures and Tables

**Figure 1 biomedicines-07-00045-f001:**
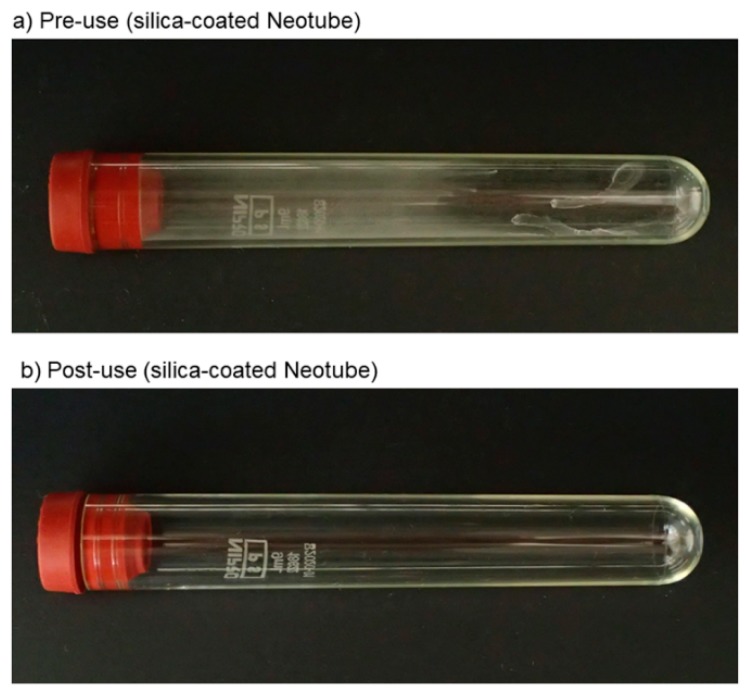
Macroscopic appearance of the silica-coated plastic tube (Neotube) pre- (**a**) and post-centrifugation (**b**).

**Figure 2 biomedicines-07-00045-f002:**
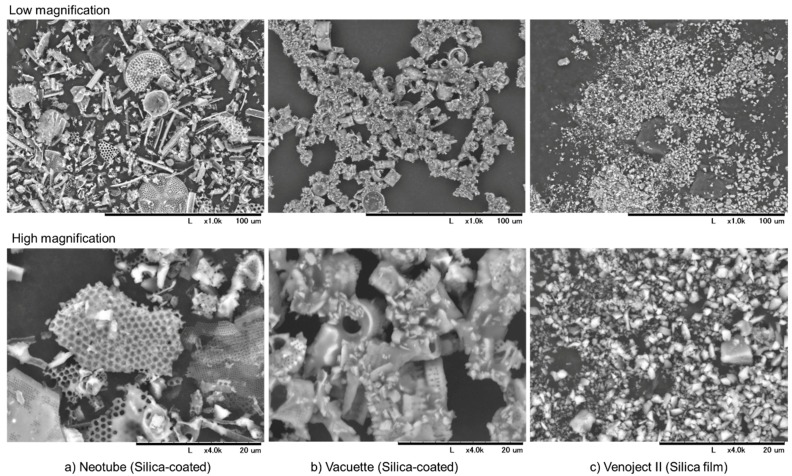
Scanning electron microscopy (SEM) observations of silica microparticles contained in (**a**) Neotube tubes (**b**) Vacuette tubes and (**c**) Venoject II tubes at low (**Upper**) and high magnification (**Lower**).

**Figure 3 biomedicines-07-00045-f003:**
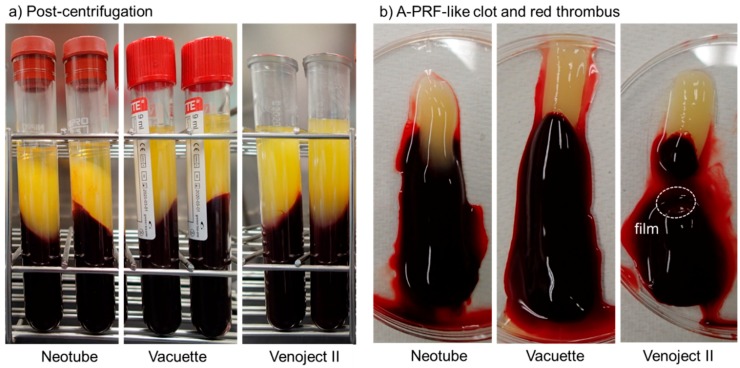
(**a**) Advanced platelet-rich fibrin (A-PRF)-like clots formed in individual plastic tubes after centrifugation and (**b**) the balance of the A-PRF-like matrix and red thrombus in 100 mm dishes.

**Figure 4 biomedicines-07-00045-f004:**
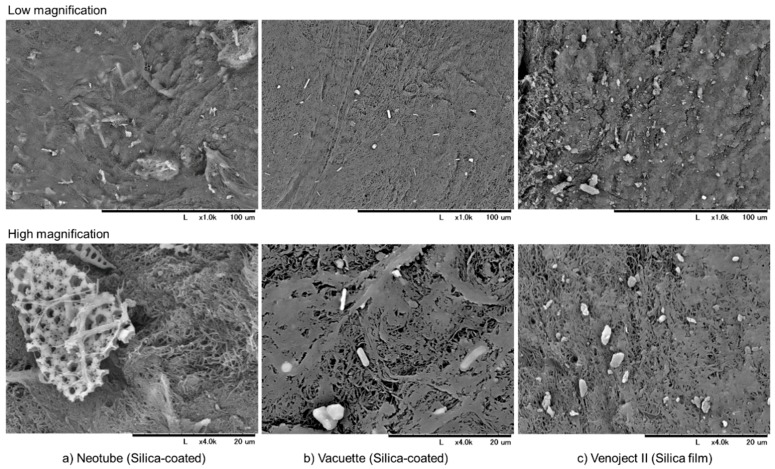
SEM observations of silica microparticles embedded in and attached to individual A-PRF-like matrices prepared by (**a**) Neotube, (**b**) Vacuette and (**c**) Venoject II.

**Figure 5 biomedicines-07-00045-f005:**
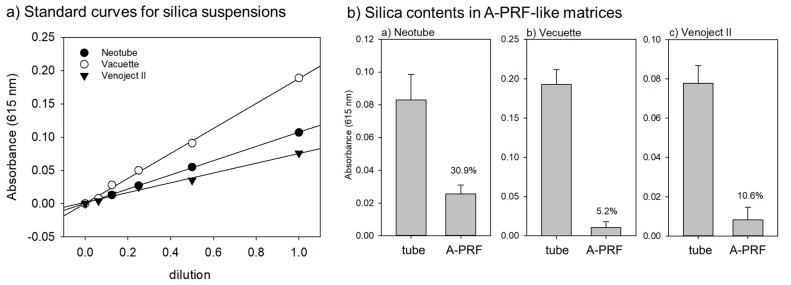
(**a**) Representative standard curves for silica contents and (**b**) contents and percentages of silica microparticles contained in A-PRF-like matrices versus whole contents of pre-use tubes. *n* = 8.

**Table 1 biomedicines-07-00045-t001:** Characteristics of plastic tubes used in this study.

	Neotube	Vacuette	Venoject II
Material of tube	plastic (PET)	plastic (PET)	plastic (PET)
Silica types	amorphous	not disclosed	not disclosed
Object coated	inner-wall surface	inner-wall surface	film
Additives	not disclosed	not disclosed	thrombin

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
