# Peer review of "Evidence for Contamination of Silica Microparticles in Advanced Platelet-Rich Fibrin Matrices Prepared Using Silica-Coated Plastic Tubes"

_biomedicines, 2019, doi:10.3390/biomedicines7020045_

Reviewer 1 Report

GENERAL COMMENTS

The manuscript is interesting, but revisions may be necessary to avoid complaints by the product manufacturers.  The authors cannot accuse the manufacturers of selling non-regulatory approved products, without providing evidence of non-approval.  I am concerned by the lack of controls used for measuring silica, and lack of comparison with glass tubes (glass also contains silica) to show what the silica concentrations differences were, if any?  It could be argued that the disinfection protocol for the plastic tubes is non-clinically relevant, since the detached silica particles will be washed away prior to tube use, and these particles would not contaminate PRF.  The introduction and discussion suffer from language problems which makes comprehension difficult.  I disagree with the publications used to support statements, the authors should ensure that the supported statements are accurate, or change the statements or publications. 

TITLE

1.    The title is acceptable.

ABSTRACT

2.    Page 1 line 26: The abstract claims that “ Such plastic tubes are approved by regulatory authorities only for diagnostic use.” This statement cannot be correct since these products are commercially available.  Please remove or provide evidence to support this statement to avoid complaints by the product manufacturers.

3.    Page 1, line 33: “both SEM examination and spectrophotometric determination demonstrated that significant levels of silica microparticles are incorporated into A-PRF-like matrix, which will be consequently incorporated into implantation sites.”  But the results showed particles and absorbance, not “significant levels” where did the significance levels results come from?

4.    Page 1, line 36: “Even though it is less toxic.” What is less toxic?  What toxicity relevance?  Toxicity was not measured, so why is it described as a result here. Please remove or revise these sentences.

5.    Page 1, line 37: “Therefore, we do not recommend usage of silica containing tubes until their safety is assured.”  But the results did not evaluate their safety.  This conclusion is not supported by the results.

INTRODUCTION

6.    Page 1, line 44. Why are 5 articles needed to support the first sentence?  These articles did not show that PRF is similar to PRP, since they focused on PRF, please remove or only cite articles that support the statement.

7.    Page 2, line 1: The meaning of this sentence is unclear and requires revision:  “This implies that fewer sections of PRF preparation protocol can be biased.”

8.    Page 2, line 4. “Due to these advantages, PRF has been increasingly used around

9.    the world.” What evidence supports this statement? PRF use has declined since the 1980’s according to some evidence.

10.  Page 2, line 10:  “If it is difficult to obtain such glass tubes, many PRF users tend to use plastic tubes specific for blood coagulation without careful quality check.”  Provide published evidence to support this statement.  Publication #6 is a review and not acceptable as evidence.

11.  Page 2, line 13:  “the lack of approval by regulatory agencies” is claimed to be supported by #6.  This is not correct, because #6 did not claim a lack of regulatory approval, and stated “criteria for their quality is not standardized.”  Please revise or provide supporting factual evidence not based on an opinion or review.

12.  Page 2, line 28:  “However, many general PRF users have rarely paid attention to their possible hazard to health.”  Please revise or provide supporting factual evidence not based on an opinion or review.

13.  Page 2, line 30:  “As these products are generally used for diagnostic purposes only, to date, manufacturers have not disclosed information regarding either specifications or ease of detachment of silica microparticles.”  Please revise or provide supporting factual evidence not based on an opinion or review.

EXPERIMENTAL SECTION

14.  Page 2, line 38:  Please describe if written informed consent was obtained from the human subjects prior to their inclusion in this study?  Please describe how the subjects were recruited?

15.  Page 3, line 3.  How is the disinfection and detachment of silica particles clinically relevant?

16.  Page 3, line 11.  Did you measure the silica content of the water or KOH? to ensure that it did not contain trace amounts of silica prior to testing?

17.  Page 5, line 10.  How did you know that the SEMs showed only silica particles and not dust or other contaminants?  Why were no controls used to show non-silica specimens?  Why were no evaluations of the particles conducted to prove they were silica?

18.  Fig 5.  How do you know the absorbance is 100% from silica and nothing else? Problem is the lack of controls.

19.  Why does the origin of Figure 5a have a negative dilution and negative absorbance?  Is negative values possible, if not, change origin to zero.

20.  Why is there no results of the statistical analyses?

21.  The results show absorbance, how does the absorbance correlate to the concentration/quantity of silica within the samples?

DISCUSSION

22.  Page 6, line 16:  “we speculate that amorphous silica is probably used.”  This is a scientific manuscript, please only describe the results or facts which can be proven, not speculated.

23.  Page 6, line 16:  “based on the general consensus regarding health hazard of crystalline silica [11,17] and/or the material safety data sheet provided by the other manufacturers [18,19]. However, we could not reach a conclusion based on this information.”  This contradicts the health hazard statements in the abstract, please ensure all statements are consistent.

24.  Page 7, line 11:  “In addition, as described in the case of complex interaction of crystalline silica [10], we cannot exclude possible interactions of amorphous silica at non-toxic levels with unidentified factors to exert unexpected health hazards, at present.”  This contradicts the health hazard statements in the abstract, please ensure all statements are consistent.

25.  Page 7, line 11:  “Until the safety of amorphous silica for implantation use is assured by international or national regulatory authorities, we should use only conventional types of plain glass tubes for PRF preparation for the benefit of patients.”  This statement contradicts the commercial sale of these products, which must have passed the safety standards.

CONCLUSIONS

26.  I disagree with the conclusion because it over-generalizes the results and discusses health hazards and the risk of complications, which were not investigated here. 

Author Response

GENERAL COMMENTS

The manuscript is interesting, but revisions may be necessary to avoid complaints by the product manufacturers.  The authors cannot accuse the manufacturers of selling non-regulatory approved products, without providing evidence of non-approval.  I am concerned by the lack of controls used for measuring silica, and lack of comparison with glass tubes (glass also contains silica) to show what the silica concentrations differences were, if any?  It could be argued that the disinfection protocol for the plastic tubes is non-clinically relevant, since the detached silica particles will be washed away prior to tube use, and these particles would not contaminate PRF.  The introduction and discussion suffer from language problems which makes comprehension difficult.  I disagree with the publications used to support statements, the authors should ensure that the supported statements are accurate, or change the statements or publications. 

Response: We thank the reviewer for the thorough review and critical comments. We understand that our thoughts do not complement. As you have mentioned, we did not perform experiments to demonstrate the possible cytotoxicity of amorphous silica microparticles in this study. However, this possibility has been frequently indicated by increasing numbers of recent publications. Thus, we thought it is sufficient to raise caution against the indiscriminate use of silica-containing blood collection tubes by clinicians. Of course, clinicians have the rights to continuously use silica-coated tubes (including tubes containing silica-coated film) for PRF preparation if they think our data are insignificant in clinical settings.

As for the control, we have repeatedly shown the SEM findings of PRF matrix prepared by glass tubes. From venipuncture to PRF preparation, all the process is done under closed conditions, which do not allow morphologically silica microparticle-like debris to enter from the outside and attach to PRF matrix. Please check the articles below if you want to check the possible existence of debris.

1)        Kitamura Y, Watanabe T, Nakamura M, Isobe K, Kawabata H, Uematsu K, Okuda K, Nakata K, Tanaka T, Kawase T*. Platelet counts in insoluble platelet-rich fibrin clots: a direct method for accurate determination. Front Bioeng Biotechnol (section Tissue Engineering and Regenerative Medicine) 6:4; 2018.

2)        Kawabata H, Isobe K, Watanabe T, Okudera T, Nakamura M, Suzuki M, Ryu J, Kitamura Y, Okudera H, Okuda K, Nakata K, Kawase T*. Quality assessment of platelet-rich fibrin-like matrix prepared from extendedly stored whole blood samples. Biomedicines 5:57; 2017.

3)        Isobe M, Suzuki M, Watanabe T, Kitamura Y, Suzuki T, Kawabata H, Nakamura M, Okudera T, Okudera H, Uematsu K, Nakata K, Tanaka T, Kawase T*. Platelet-rich fibrin prepared from stored whole-blood samples. Int J Implant Dent 3:6; 2017.

As for your comment “It could be argued that the disinfection protocol for the plastic tubes is non-clinically relevant,” we are sorry that we did not understand what you meant. We wonder why the evacuated blood collection tubes need to be washed prior to use. It seems impossible for us to wash the inside of silica-coated tubes without sacrificing their vacuum state.

English was again edited by a professional service.

TITLE

1.    The title is acceptable.

ABSTRACT

2.    Page 1 line 26: The abstract claims that “ Such plastic tubes are approved by regulatory authorities only for diagnostic use.” This statement cannot be correct since these products are commercially available.  Please remove or provide evidence to support this statement to avoid complaints by the product manufacturers.

Response: In this study, we used silica-containing plastic tubes that are approved as diagnostic tubes in Japan or Europe and are commercially available as a medical device. Furthermore, the manufacturers have issued repeated warnings, like “do not use for the preparation of something like PRF” because some dentists ignore such a warning and use them based on their discretion, as it is allowed by the regulation in Japan.

Unfortunately, we do not know much about the Intra-Spin tubes. However, to get those products approved for therapeutic use (e.g., for the preparation of PRF), bone substitutes may be required to be examined by stricter testing to qualify as a class III medical device. We are confident that the manufacturers will not complain against our statement.

3.    Page 1, line 33: “both SEM examination and spectrophotometric determination demonstrated that significant levels of silica microparticles are incorporated into A-PRF-like matrix, which will be consequently incorporated into implantation sites.”  But the results showed particles and absorbance, not “significant levels” where did the significance levels results come from?

Response: Based on the high reproducibility, we used the term “significant.” In the control PRF matrix prepared by glass tubes, there is no silica microparticles incorporated into PRF matrix. It is because glass tubes do not produce silica microparticles during the process of PRF preparation. Thus, it is “significant” vs. none.

Since we did not examine the possible incorporation into implantation sites or the possible toxicity of the implanted silica microparticles, we did not intend to refer to the “significance” in the possible health hazards.

4.    Page 1, line 36: “Even though it is less toxic.” What is less toxic?  What toxicity relevance?  Toxicity was not measured, so why is it described as a result here. Please remove or revise these sentences.

Response: It has been generally thought that crystalline silica is more toxic than amorphous silica. However, it does not imply that amorphous silica is not toxic, but it is safe enough to be implanted in our body. In contrast, recent publications have demonstrated that amorphous silica is less toxic but not safe. This is our explanation for using “less toxic.” However, to address the comments made by reviewer #2, we have extensively re-written this part to emphasize the potential health hazards.

5.    Page 1, line 37: “Therefore, we do not recommend usage of silica containing tubes until their safety is assured.”  But the results did not evaluate their safety.  This conclusion is not supported by the results.

Response: We know the fact that many clinicians use silica-containing tubes without the awareness about the contamination of silica microparticles derived from plastic tubes. Here, we have demonstrated the contamination of silica microparticles in the PRF matrix. This finding is good enough to caution the uninformed “innocent” clinicians. However, according to the comments made by reviewer #2, we have extensively re-written this part to emphasize the potential health hazards.

 INTRODUCTION

6.    Page 1, line 44. Why are 5 articles needed to support the first sentence?  These articles did not show that PRF is similar to PRP, since they focused on PRF, please remove or only cite articles that support the statement.

Response: In this sentence, we meant that PRF is categorized in a PRP family and that the major mechanism of PRF action can be identified as the concentrated growth factors stored in platelets as is that of conventional PRP. Thus, we believe these citations are acceptable for this statement.

7.    Page 2, line 1: The meaning of this sentence is unclear and requires revision:  “This implies that fewer sections of PRF preparation protocol can be biased.”

Response: This sentence means that the protocol for PRF preparation is composed of fewer sections; this protocol is less biased than the conventional, manual preparation protocol of PRP.

8.    Page 2, line 4. “Due to these advantages, PRF has been increasingly used around    the world.” What evidence supports this statement? PRF use has declined since the 1980’s according to some evidence.

Response: To our knowledge, a series of the first papers of Choukroun’s original PRF (L-PRF) was published by you in the middle of 2000s. Thereafter, some modifications have been made on the original PRF preparation protocol to create PRF derivatives, such as A-PRF. In parallel, the number of PRF users has been increasing to date. This phenomenon can be endorsed by the increased sales of their specific centrifuges, including a Medifuge centrifuge. We had obtained this information from the distributor of Medifuge in Japan.

Thus, we do not know about the occurrences of the 1980s. It was much earlier than the appearance of the first PRF report.

10.  Page 2, line 10:  “If it is difficult to obtain such glass tubes, many PRF users tend to use plastic tubes specific for blood coagulation without careful quality check.”  Provide published evidence to support this statement.  Publication #6 is a review and not acceptable as evidence.

Response: To our knowledge, there are no publications supporting the statement; however, at least in Japan, according to the distributor’s advice, many CGF users have used silica-containing plastic tubes or glass tubes made in China, which are imported as a device for chemical experiments.

In our previous article #6, we described in detail the quality of medical devices, which may influence the quality of PRF, to raise awareness among clinicians. Thus, we provided the source of information to help the readers understand the meaning of our statement.

11.  Page 2, line 13:  “the lack of approval by regulatory agencies” is claimed to be supported by #6.  This is not correct, because #6 did not claim a lack of regulatory approval, and stated “criteria for their quality is not standardized.”  Please revise or provide supporting factual evidence not based on an opinion or review.

Response: We request you to carefully read our previous article #6 again. We meant the quality of platelet concentrates by “criteria for their quality is not standardized.” In contrast, we have not mentioned the quality of blood collection tubes for PRF preparation in this part, although Article #6 briefly described the background of the approval system of medical devices in Japan.

12.  Page 2, line 28:  “However, many general PRF users have rarely paid attention to their possible hazard to health.”  Please revise or provide supporting factual evidence not based on an opinion or review.

Response: We understand the essential rule that opinion based on personal subjectivity should be avoided in scientific papers. However, we think that this principle could be applied mainly to scientific matter, although not advisable for all matters. Several non-scientific, economical, or social matters cannot be always endorsed by published data sheets. If this kind of information is correct, but not distorted or misevaluated, and does not influence biomedical data analysis or interpretation, we think such information is acceptable.

So far, we have interviewed many dentists, who are CGF users invited by a certain distributor in Japan, and they use only the information provided by the distributor in their decision-making in choosing blood collection tubes. We have described this part based on this information.

13.  Page 2, line 30:  “As these products are generally used for diagnostic purposes only, to date, manufacturers have not disclosed information regarding either specifications or ease of detachment of silica microparticles.”  Please revise or provide supporting factual evidence not based on an opinion or review.

Response: The manufacturers have not disclosed such information. We have only mentioned a fact.

We solicited the information from the manufacturers, but sought reply from only one company, NIPRO, which stated that amorphous silica, celite, is being used in their product, but the company did not provide any additional information regarding the purity or ease of detachment of celite.

EXPERIMENTAL SECTION

14.  Page 2, line 38:  Please describe if written informed consent was obtained from the human subjects prior to their inclusion in this study?  Please describe how the subjects were recruited?

Response: We have added a note that we obtained written informed consent from the human subjects prior to blood collection. Blood samples were donated by the members of our research groups and their family members.

15.  Page 3, line 3.  How is the disinfection and detachment of silica particles clinically relevant?

Response: We did not understand what you meant in this comment. However, silica microparticles used for coating must be sterilized to be approved as a medical device. As for the detachment of silica microparticles, in addition to silica’s cytotoxicity, the degree of contaminants in silica particles is unknown. Thus, as we have repeatedly described in the text, it is possible that silica microparticles implanted with PRF into the implantation sites may cause tumors or inflammation that would hamper tissue regeneration or repair.

16.  Page 3, line 11.  Did you measure the silica content of the water or KOH? to ensure that it did not contain trace amounts of silica prior to testing?

Response: After digestion with KOH, samples were washed and resuspended in milli Q water. We used pure water (or KOH) as the blank sample. Thus, even though pure water (or KOH) contains trace amounts of silica microparticles or nanoparticles, those contaminants can be subtracted.

17.  Page 5, line 10.  How did you know that the SEMs showed only silica particles and not dust or other contaminants?  Why were no controls used to show non-silica specimens?  Why were no evaluations of the particles conducted to prove they were silica?

Response: As SEM findings are used to demonstrate blood cell distribution, we examined SEM findings of PRF prepared by plain glass tubes and compared with that prepared by silica-coated tubes. We did not show non-silica controls (please see the references listed in our response to your general comment), but we can easily identify most of those artificial materials by comparing with the SEM findings of silica microparticles detached from the inner walls or films.

In addition, to support this finding, we developed and performed the quantitative spectrophotometric assay of silica microparticles, as shown in Figure 5.

18.  Fig 5.  How do you know the absorbance is 100% from silica and nothing else? Problem is the lack of controls.

Response: It is a basic technique to use a blank sample and set the absorbance to zero.

19.  Why does the origin of Figure 5a have a negative dilution and negative absorbance?  Is negative values possible, if not, change origin to zero.

  Response: It is a linear regression. The negative value is just theoretically possible, but not actually possible.

20.  Why is there no results of the statistical analyses?

Response: We did not think it is necessary to compare those data and perform statistical analyses. We did not intend to demonstrate any biomedical significance. The purpose of these data was to show that silica microparticles are possibly included in the PRF matrix. In our evaluation, the levels of these contaminants were much higher than “accidental.”

21.  The results show absorbance, how does the absorbance correlate to the concentration/quantity of silica within the samples?

Response: Please find the left linear regression curves. A dilution of 1.0 means full amounts of silica microparticles in individual tubes.

DISCUSSION

22.  Page 6, line 16:  “we speculate that amorphous silica is probably used.”  This is a scientific manuscript, please only describe the results or facts which can be proven, not speculated.

Response: From the specific appearance of the diatomaceous earth, it is possible to identify the microparticles, amorphous silica 99% of the times. In addition, crystalline silica is known to be more toxic and cause various diseases, while amorphous silica has been used as an additive in various products, including tooth paste and processed foods. Thus, we are sure that this speculation is justifiable in a scientific paper.

23.  Page 6, line 16:  “based on the general consensus regarding health hazard of crystalline silica [11,17] and/or the material safety data sheet provided by the other manufacturers [18,19]. However, we could not reach a conclusion based on this information.”  This contradicts the health hazard statements in the abstract, please ensure all statements are consistent.

Response: As described in our earlier responses, we think that the silica used for coating is an amorphous type. In fact, Neotube uses amorphous silica. For other products, we believe the silica is also of the amorphous type, but the disclosed information is not sufficient to identify if it is amorphous silica. It is not related to the health hazard statements in the abstract.

24.  Page 7, line 11:  “In addition, as described in the case of complex interaction of crystalline silica [10], we cannot exclude possible interactions of amorphous silica at non-toxic levels with unidentified factors to exert unexpected health hazards, at present.”  This contradicts the health hazard statements in the abstract, please ensure all statements are consistent.

Response: We just mentioned the possibility of further health hazard of amorphous silica. In the abstract, because of the word count limitation, we had to keep the text concise. However, in the Discussion section, we are expected to expand the possibility to state our purpose. We think there are no contradictions between the abstract and the discussion.

25.  Page 7, line 11:  “Until the safety of amorphous silica for implantation use is assured by international or national regulatory authorities, we should use only conventional types of plain glass tubes for PRF preparation for the benefit of patients.”  This statement contradicts the commercial sale of these products, which must have passed the safety standards.

Response: We do not know the case of Intra-Spin plastic tubes. In case of most blood collection tubes, however, their safety is approved as a device for diagnostic use, but not as an implantation material. In addition, the manufacturers we mentioned in this study mention “diagnostic use only.” The clinicians who use these products should consider their responsibility to their patients. Thus, we don’t think this statement contradicts any commercial sales.

CONCLUSIONS

26.  I disagree with the conclusion because it over-generalizes the results and discusses health hazards and the risk of complications, which were not investigated here.

Response: We just suggested the possible toxicity, which has been demonstrated by various experimental systems. We do not think it is wrong. At the same time, we do not think it is beneficial for anybody to pursue this matter to identify its toxicity, because the manufacturers actually caution the clinicians regarding the use of silica-coated products in PRF preparation. We just suggest such clinicians to reconsider their use of silica-coated tubes, including tubes containing silica-coated film.

Please see Becton-Dickinson’s MSDS below, which indicates hazards identification of silica, for example.

Reviewer 2 Report

The present study titled: “Evidence for contamination of silica microparticles in advanced platelet-rich fibrin matrix prepared using silica-coated plastic tubes” is a very interesting study that this reviewer greatly suggests publishing. In fact, many clinicians have been complaining of issues related to additional inflammation by several tubes and this offers a perfect explanation. I would rank this as a top 5% of all papers published on the topic. I only have minor comments as highlighted below.

1) In the abstract: It is written: “… regenerative dentistry, requires plain glass tubes…” Actually to get clotting you do not ‘REQUIRE’ glass tubes. It is certainly optimized but not required. I would simply write ‘optimized’.

2) In the abstract: “ Such plastic tubes are approved by regulatory authorities only for diagnostic use.” Actually these reviewers need to be better informed and this reviewer wants to clarify the situation in North America and Europe. In fact glass tubes are NOT cleared because of the potential risk of ‘breaking them’ with blood samples posing a health hazard. Therefore, the only FDA-cleared tubes for the production of PRF are silica-based plastic tubes. This should be better reported throughout the article. The authors make it seem like the glass tubes are improved and the silica-coated ones are not.

That being said, this reviewer fully and entirely supports the findings from this study. This reviewer also feels the glass tubes are much better and that the silica tubes – specifically the A-PRF tubes are causing massive inflammation in patients. I have therefore advised the editorial board to rapidly publish this paper online.

3) The concluding sentence needs to be reformulated in the abstract: “ Even though it is less toxic, we cannot exclude the possibility that silica microparticles negatively influence tissue regeneration. Therefore, we do not recommend usage of silica-containing tubes until their safety is assured.” You don’t know if it is more or less toxic in fact. I would simply state to the facts such as “The production of A-PRF utilizing silica-coated tubes produce a much greater inflammatory response when compared to pure glass tubes” (or something equivalent). Please stick to facts.

4) The authors need to change the following sentence because it is inaccurate “ Therefore, some PRF users purchase PRF-specific glass tubes, even though conventional glass tubes are less expensive and can be obtained routinely without authorized distributors.” In fact again, conventional glass tubes are far more expensive than plastic coated tubes with silica. In the united states, these plastic tubes are about half the cost. That’s the first issue. The second issue again – a clinician CANNOT purchase plain glass tubes and utilized them for a clinical procedure. Therefore, these authors have not an idea of the seriousness of utilizing non-FDA cleared system. This reviewer is a University based professor unable to use plain glass tubes despite knowing full well they work more effectively and are safer for patients owing to the FDA restricting only FDA-cleared equipment. Therefore, this manuscript is very important because it will show the FDA and CE the importance of utilizing glass. But this reviewer worries that with the number of mistakes and lack of understanding of the regulatory situation across the world that the manuscript will not be taken seriously. Please correct.

5) For the following sentence: “ medical and dental clinicians in Japan have rights to choose those products on basis of their discretion in the treatment of patients [6]. This reviewer again would appreciate if the authors can think about the manuscript globally. This is important for the entire world and this statement should be modified accordingly. This reviewer recommends that the authors think much bigger. This is a very important study.

6) Have the authors tested the A-PRF tubes by Process for PRF. They have so much inflammation and silica on the walls. That would be a great one to add to this study.

Author Response

The present study titled: “Evidence for contamination of silica microparticles in advanced platelet-rich fibrin matrix prepared using silica-coated plastic tubes” is a very interesting study that this reviewer greatly suggests publishing. In fact, many clinicians have been complaining of issues related to additional inflammation by several tubes and this offers a perfect explanation. I would rank this as a top 5% of all papers published on the topic. I only have minor comments as highlighted below.

1) In the abstract: It is written: “… regenerative dentistry, requires plain glass tubes…” Actually to get clotting you do not ‘REQUIRE’ glass tubes. It is certainly optimized but not required. I would simply write ‘optimized’.

Response: Thank you for your advice. We have modified the sentence.

2) In the abstract: “ Such plastic tubes are approved by regulatory authorities only for diagnostic use.” Actually these reviewers need to be better informed and this reviewer wants to clarify the situation in North America and Europe. In fact glass tubes are NOT cleared because of the potential risk of ‘breaking them’ with blood samples posing a health hazard. Therefore, the only FDA-cleared tubes for the production of PRF are silica-based plastic tubes. This should be better reported throughout the article. The authors make it seem like the glass tubes are improved and the silica-coated ones are not.

Response: We agree with you. Indeed, we also want to survey real situations of usage of PRF tubes at global levels. At present, unfortunately, what we can do is limited to information collection through Internet: in most cases, such information is obtained from private web sites of manufacturers or distributors. Thus, although we know it is risky to believe all of what is described there, we have to pick some reliable information for our research activity.

Because of the limited information, we cannot identify the FDA-cleared silica-coated plastic tubes for PRF production. As far as we know, according to the manufacture’s web site, the Intra-Spin’s “Yellow Tube” is the only FDA-registered one. However, it also mentions “Our PRF Tubes are plain lab tubes in a sterile package and are used off-label in operating rooms, hospitals and all cases where full sterilization is mandatory and when the blood collected is going to be processed and used back into the patient’s body, like in the process of growth factors in PRF and PRP (off label use).” On the other hand, the Intra-Spin’s “PRF red tube” is an FDA-registered, plain glass tube. According to the statement on the web site, this tube can be used off-label for PRF preparation. Please see the links below.

http://bocadentalsupply.com/prf-tubes-yellow-liquid-prf-off-label-plastic-10-ml-no-additive-100-tubes-per-box/

http://bocadentalsupply.com/prf-tubes-red-prf-membranes-and-plugs-glass-10-ml-no-additive-100-tubes-per-box/

Thus, we understand that the use of these tubes for PRF preparation is off-label use. If our understanding is correct, these tubes are registered just as blood-collection tubes. Unfortunately, we need more time to thoroughly investigate this matter and collect reliable information that can be added to the text. However, according to your advice, we modified some expressions regarding plastic tubes.

3) The concluding sentence needs to be reformulated in the abstract: “ Even though it is less toxic, we cannot exclude the possibility that silica microparticles negatively influence tissue regeneration. Therefore, we do not recommend usage of silica-containing tubes until their safety is assured.” You don’t know if it is more or less toxic in fact. I would simply state to the facts such as “The production of A-PRF utilizing silica-coated tubes produce a much greater inflammatory response when compared to pure glass tubes” (or something equivalent). Please stick to facts.

Response: Thank you for your advice. We rewrote this part to emphasize the potential risks of the silica-coated tubes.

4) The authors need to change the following sentence because it is inaccurate “ Therefore, some PRF users purchase PRF-specific glass tubes, even though conventional glass tubes are less expensive and can be obtained routinely without authorized distributors.” In fact again, conventional glass tubes are far more expensive than plastic coated tubes with silica. In the united states, these plastic tubes are about half the cost. That’s the first issue. The second issue again – a clinician CANNOT purchase plain glass tubes and utilized them for a clinical procedure. Therefore, these authors have not an idea of the seriousness of utilizing non-FDA cleared system. This reviewer is a University based professor unable to use plain glass tubes despite knowing full well they work more effectively and are safer for patients owing to the FDA restricting only FDA-cleared equipment. Therefore, this manuscript is very important because it will show the FDA and CE the importance of utilizing glass. But this reviewer worries that with the number of mistakes and lack of understanding of the regulatory situation across the world that the manuscript will not be taken seriously. Please correct.

Response: In this sentence, we intended to draw the attention of the readers toward the import of non-approved glass tubes from China, which had been produced for experimental use, in Japan. These products are actually cheaper than the approved plastic tubes, and the distributor sells them to clinicians for PRF preparation with a caution “for experimental use only.” Thus, we modified the sentence by adding “non-approved.”

As for the second issue mentioned by you, unfortunately, we only have a superficial knowledge of the situations in the United States. If you are right, which we think you are, we could understand by your explanation why the US regulatory agency prohibits the use of glass tubes in clinical settings. However, it is difficult for us to understand why some distributors keep selling glass tubes produced by Intra-Spin or PRF Process under these situations. For example, a Taiwanese trading company, Global Sources, sells glass tubes approved by CE and ISO, although we think both CE and ISO approve their standard operating procedure, but not their quality. Please see the link below.

https://m.globalsources.com/si/AS/ANJN-International/6008850741268/pdtl/Vacuum-blood-collection-tube/1132544713.htm

We agree with your idea about the necessity of glass tubes for PRF preparation. At the same time, we think we need to globally survey the situation of glass tubes for medical use by international collaboration.

5) For the following sentence: “ medical and dental clinicians in Japan have rights to choose those products on basis of their discretion in the treatment of patients [6]. This reviewer again would appreciate if the authors can think about the manuscript globally. This is important for the entire world and this statement should be modified accordingly. This reviewer recommends that the authors think much bigger. This is a very important study.

Response: We have re-written this sentence so that the readers can easily comprehend the intended meaning. Since we do not know the situation in individual countries’, we could not expand this paragraph deeply. Instead, we have proposed international collaboration to survey the current situation of the individual countries and establish universal standards for patient-oriented, but not business-centered, better PRF therapy.

6) Have the authors tested the A-PRF tubes by Process for PRF. They have so much inflammation and silica on the walls. That would be a great one to add to this study.

Response: We have sometimes used glass tubes made by Process for PRF, mainly in our pre-clinical studies. Some colleagues have used these glass tubes in clinical settings, but they have never found serious complications, like severe inflammation. Thus, unfortunately, we cannot add anything like that.